# An electrically controlled single-molecule spin switch

Wantong Huang [1,6], Kwan Ho Au-Yeung [1,2,6], Paul Greule [1], Máté Stark[1], Christoph Sürgers [1], Wolfgang Wernsdorfer [1,3], Roberto Robles [4], Nicolas Lorente [4,5] & Philip Willke [1,2] ✉

Precise control of spin states and spin-spin interactions in atomic-scale magnetic structures is crucial for spin-based quantum technologies. A promising architecture is molecular spin systems, which offer chemical tunability and scalability for larger structures. An essential component, in addition to the qubits themselves, is switchable qubit-qubit interactions that can be individually addressed. In this study, we present an electrically controlled single-molecule spin switch based on a bistable complex adsorbed on an insulating magnesium oxide film. The complex, which consists of an Fe adatom coupled to an iron phthalocyanine (FePc) molecule, can be reversibly switched between two stable states using bias voltage pulses locally via the tip of a scanning tunnelling microscope. Inelastic electron tunnelling spectroscopy measurements and density functional theory calculations reveal a distinct change between a paramagnetic and a non-magnetic spin configuration. Lastly, we demonstrate the functionality of this molecular spin switch by using it to modify the electron spin resonance frequency of a nearby target FePc spin within a spin-spin coupled structure. Thus, we showcase how individual molecular machines can be utilized to create scalable and tunable quantum devices.

Individual electronic spins constitute promising building blocks for quantum information processing, quantum simulation, and quantum sensing. For that, several platforms are currently being explored that aim to harness spin qubit systems for quantum technologies[1], including molecular spin systems[2–5]: They are an appealing class, since they can be chemically designed and efficiently scaled up through self-assembly in long-range ordered structures. Towards the realization of functional quantum devices, not only the basic qubit units required, but also a wide spectrum of functional units such as auxiliary qubits, electric control and tunable interqubit structures[5].

In this context, switchable qubit-qubit interactions are crucial for the implementation of multi-qubit gates: They allow one to

rapidly control the interactions between individual qubits. These have been explored for molecular spins in both experiment and theory by a variety of approaches, including global microwave (MW) pulses on different qubits[6,7] or on spin switches placed between them[8–10]. The latter approach, however, requires a local spin switch that can be altered either coherently or incoherently on a fast timescale. Moreover, these realizations still relied on ensemble electron spin resonance (ESR) experiments that do not grant access to a local control of individual qubits. In that regard, one realization was theoretically proposed, in which the tip of a scanning tunnelling microscope (STM) is used to implement two-qubit gates in polyoxometalate molecules: Here, two localized spins ($S = ½$) within the

[1]Physikalisches Institut, Karlsruhe Institute of Technology, Karlsruhe, Germany. [2]Center for Integrated Quantum Science and Technology (IQST), Karlsruhe Institute of Technology, Karlsruhe, Germany. [3]Institute for Quantum Materials and Technologies, Karlsruhe, Germany. [4]Centro de Física de Materiales CFM/MPC (CSIC-UPV/EHU), Donostia-San Sebastián, Spain. [5]Donostia International Physics Center, Donostia-San Sebastián, Spain. [6]These authors contributed equally: Wantong Huang, Kwan Ho Au-Yeung. ✉e-mail: philip.willke@kit.edu

molecule can be coupled by injecting a tunnelling electron into the molecule's central core[11].

Single-molecule switches, as part of the framework of synthetic molecular systems coined artificial molecular machines[12], offer an alternative degree of freedom (mechanical, electronic or magnetic) as well as bistability that could potentially tune qubit-qubit interactions. They enable reversible transitions between stable states in response to external stimuli[12]: If integrated into spin circuits, they could control magnetic interactions between spin-containing molecules. For example, spin-crossover molecules[13] demonstrate switchable spin states via e.g. electric fields, light or inelastic scattering with tunnelling electrons.

To probe molecular spins at the atomic scale, low-temperature STM is an excellent technique, since it can precisely manipulate and characterize them one-by-one. Recent advances combining ESR with STM[14,15], offering high energy resolution of several MHz (tens of neV), have allowed to detect magnetic coupling between atomic spins[16] and to perform coherent control on single and multi-spin systems[17,18]. This approach has since been extended to molecular spin systems[19–24]. Using ESR-STM, magnetic switches based on rare-earth atoms such as Dy[25] and Ho[26] have demonstrated local tuning of the resonance frequency of nearby spins. However, their stability remains limited by the onset of diffusion at elevated temperatures[27] and they require direct interaction with inelastic tunnelling electrons for the change of their magnetic state. These challenges motivate the development of a spin switch within a molecular framework. In return, while bistable molecular structures have been extensively investigated using STM[28–35], none have yet demonstrated the ability to reversibly tune the magnetic state of a neighbouring spin.

In this work, we investigate and implement molecular spin switches, electrically controlled via short bias voltage pulses from an STM tip. We construct these switches from complexes that consist of single Fe adatoms and FePc molecules, via tip-assisted on-surface assembly. Density functional theory (DFT) calculations of the two switching configurations highlight the essential role of the Fe atom in reshaping the energy landscape and thereby enabling bistability. Combined inelastic electron tunnelling spectroscopy (IETS) measurements and DFT calculations reveal a reversible change in the molecular spin states, switching between configurations, $S > 0$ and $S = 0$. We

demonstrate the functionality of the spin switch by tuning the resonance frequency of a nearby FePc target spin center, which is magnetically coupled to a switch. Here, we utilize ESR-STM and its high energy resolution (~neV) to i) directly detect the change in the magnetic dipole field of the switch and ii) resolve the weak intermolecular magnetic coupling. As ESR is compatible with coherent spin control[20], our system serves as a proof-of-concept device for reversible, switchable qubit-qubit interactions within a molecule-based quantum platform. Bridging the fields of molecular machines, local bottom-up assembly as well as spin-based quantum control, this work provides a foundational step towards a scalable molecular quantum architecture.

## Results and discussion

The experiments were performed in a low-temperature STM with a base temperature of ~50 mK. Figure 1a shows a topographic image of the sample consisting of self-assembled molecular islands of pristine FePc along with individual Fe adatoms. FePc has been shown to be a mostly isotropic spin $S = 1/2$ system when adsorbed on MgO/Ag(001)[19,20,24], while individual Fe adatoms are a spin $S = 2$ with an out-of-plane magnetic anisotropy barrier $D = -4.6$ meV[36].

Previously, it was demonstrated that stable Fe-FePc complexes can be formed by STM tip manipulation: Here, the Fe adatom is located directly underneath one of the benzene ligands of FePc, forming an $Fe(C_6H_6)$ half-sandwich complex that results in a reduction of the Fe atom spin state[24,37]. In this study, we find that the same strategy, i.e. employing STM vertical manipulation, can be used to realize a bistable single molecule switch (Fig. 1b) when their relative alignment is slightly different: After picking up a single FePc and releasing it onto an Fe adatom on 2 ML MgO/Ag(001), the subsequent STM image (Fig. 1c) shows that the Fe-FePc complex has a similar cross-like appearance to the pristine FePc, but this complex (assigned as State A) exhibits a higher apparent height than the pristine one (Supplementary Note 1).

After applying a short STM voltage pulse above the complex (~3 s), the subsequent STM image shows that the Fe-FePc complex appears in a different contrast (Fig. 1d). The molecule shows clearly different features both in the center and on the ligands, and it has a lower overall apparent height of 70 pm compared to State A (Supplementary Note 1). Moreover, two of the four isoindole ligands are distorted,

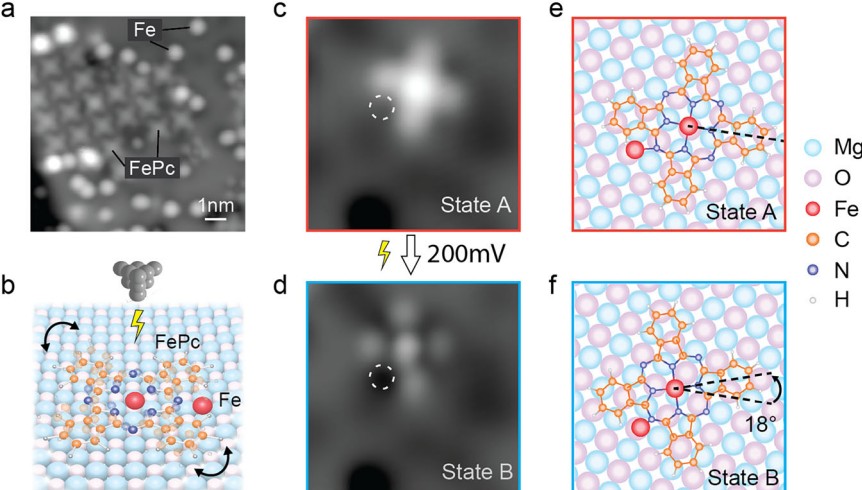

**Fig. 1 | Adsorption and bistability of Fe-FePc complexes on 2 ML MgO/Ag(001).**
**a** Overview STM image (10 nm × 10 nm; $V_{DC} = -200$ mV, $I = 60$ pA) showing a self-assembled molecular island of pristine FePc molecules, and single Fe adatoms on 2 ML MgO/Ag(001). **b** Schematic of a switchable Fe-FePc complex controlled by a voltage pulse from the STM tip. STM images (3 nm × 3 nm) of the complex (**c**) before and (**d**) after applying a short bias voltage pulse ($V_{DC} = 200$ mV, $I = 20$ pA) over the molecule center. ($I = 20$ pA, **c**: $V_{DC} = -100$ mV, **d**: $V_{DC} = 100$ mV). The

complex appears in a "bright" (State A) or "dark" (State B) contrast depending on its state. The Fe adatom position is indicated by a dashed circle. **e, f** Adsorption geometries (top view) obtained from DFT calculations of the Fe-FePc complex in **e** State A and **f** State B. In both cases, the Fe adatom is located in good approximation on an oxygen binding site of MgO, while FePc is located close to a Mg site in State B. The two configurations are rotated by 18 degrees. Red and blue frames in **c**–**f** correspond to State A and State B, respectively.

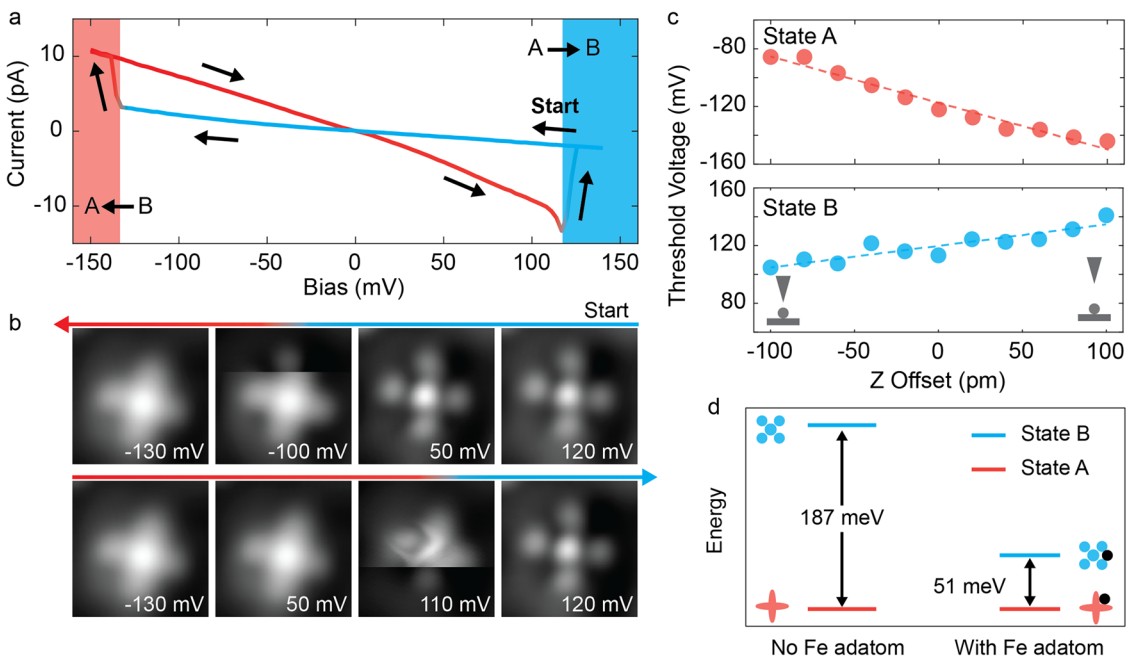

**Fig. 2 | STM-induced reversible switching of the Fe-FePc complex. a** The $I(V)$ spectrum taken at the center of the Fe-FePc complex shows the switching between the two states, which additionally reveals a hysteresis behavior. Voltage sweep directions of decreasing and increasing bias are indicated by the arrows, starting from the positive side. The red and blue areas indicate the threshold voltage for switching between States A and B, respectively. **b** STM images (2 nm × 2 nm; $I$ = 20 pA; scanning from top to bottom with forward and backward movement) taken at different biases that pinpoint the switching events. The colored arrows mark the scanning order. The scan starts at 120 mV in State B, switches to State A during the pass at −100 mV, and returns to State B at 110 mV. **c** Threshold voltage dependence on Z offset for State A (top) and State B (bottom). The feedback loop opens at zero Z-offset ($V_{DC}$ = −130 mV, $I$ = 20 pA, raw datasets are shown in Supplementary Note 8). Dashed lines represent linear fits, indicative of electric field-driven switching with slopes of −0.32 mV/pm for A, and 0.14 mV/pm for B, respectively. The sketches illustrate the proximity of the tip. **d** Comparison of absolute energies between State A and B with and without Fe adatom from DFT calculations. Without the Fe adatom, the energy difference is 187 meV, whereas with Fe, it is reduced to 51 meV.

together with an additional dark contrast in between, indicating the position of the Fe adatom. We assign this Fe-FePc complex as State B, which is compared to State A rotated by about 18° around the Fe adatom. Based on a lattice site analysis (Supplementary Note 2) and the DFT-calculated adsorption geometries (Fig. 1e, f and Supplementary Note 3), we conclude that, in both cases, the Fe adatom is situated between two isoindole ligands, but depending on the state, the alignment and lattice site is slightly different: for State A, both the Fe adatom and the FePc molecule center are adsorbed on an oxygen binding site of MgO, which is also the case for pristine Fe and FePc[19,38]. In contrast, for State B, the FePc center is close to a Mg site of the MgO lattice. In both cases, the FePc center and Fe adatom are in close proximity: the MgO lattice difference between them is around (2, 0) for State A, and around (1.5, 0.5) for State B. Crucially, we find that the switching between the two states is highly controllable when applying bias voltage pulses (Supplementary Note 4).

In order to shed light on the switching mechanism between the two states, we perform $I(V)$ spectra. Figure 2a shows a hysteresis behavior when sweeping the bias voltage across both polarities above the Fe-FePc complex. Together with the bias-dependent STM images (Fig. 2b), these results indicate that the switching of the bistable conformation is bias polarity dependent. The switching threshold varies in the range of $V_{DC}$ = ± (60 − 250) mV among different Fe-FePc complexes (Supplementary Note 5), possibly due to different tip geometries and local adsorption environments of the complex. In this specific case shown in Fig. 2a, the threshold voltage for B → A is at −133 mV and for A → B at +117 mV, respectively. For even higher bias voltages beyond the threshold voltage, we do not observe any switching between the two states. Instead, we observe a gradual tendency towards random rotation within a given state (Supplementary Note 6 and Supplementary Note 7). In Fig. 2c, we

plot the threshold voltage as a function of tip height (Supplementary Note 8 for raw $I(V)$ spectra), which reveals a linear relation $V_{thresh} \propto \Delta z$. This behavior would be in alignment with an electric field driven switching mechanism as found elsewhere for molecular systems on a surface[28,39–43]. It also agrees with the observations of a nearly immediate response of the switching process, in contrast to what is expected for switching induced by rate-dependent inelastic tunneling events[44,45]. Due to the small threshold voltages found here (-100 mV), we cannot fully exclude the contribution from inelastic tunneling electrons for overcoming the potential energy barrier or a combination of both[46–49]. However, we note that the asymmetry in the stability with bias voltage is usually not expected from inelastic tunneling electron excitations. Moreover, we find that the switching is often also feasible when the tip is not located directly on top of the complex, but up to around a nanometer to the side. From the theoretical side, we find that, by employing DFT calculations, the difference in binding energy between State A and B is 51 meV (Fig. 2d and Supplementary Note 3). This shows that both states are very close in energy and supports our observation that they could easily be switched via the STM tip. In contrast, for pristine FePc, where the Fe adatom is absent, the energy difference between State A and B is 187 meV, and the controllable switch is not observed, suggesting that the Fe adatom is crucial for enabling the bistability.

To probe the magnetic properties of our spin systems, we perform $dI/dV$ spectroscopy measurements on the complex in both configurations. Figure 3a shows a $dI/dV$ spectrum measured at the FePc center of the complex in State A, revealing several symmetric step features. We attribute these steps to inelastic electron tunneling spectroscopy (IETS) excitations from the magnetic ground state to the excited spin states. The magnetic nature of the IETS features is further supported by setpoint-dependent $dI/dV$ measurements with a magnetic tip

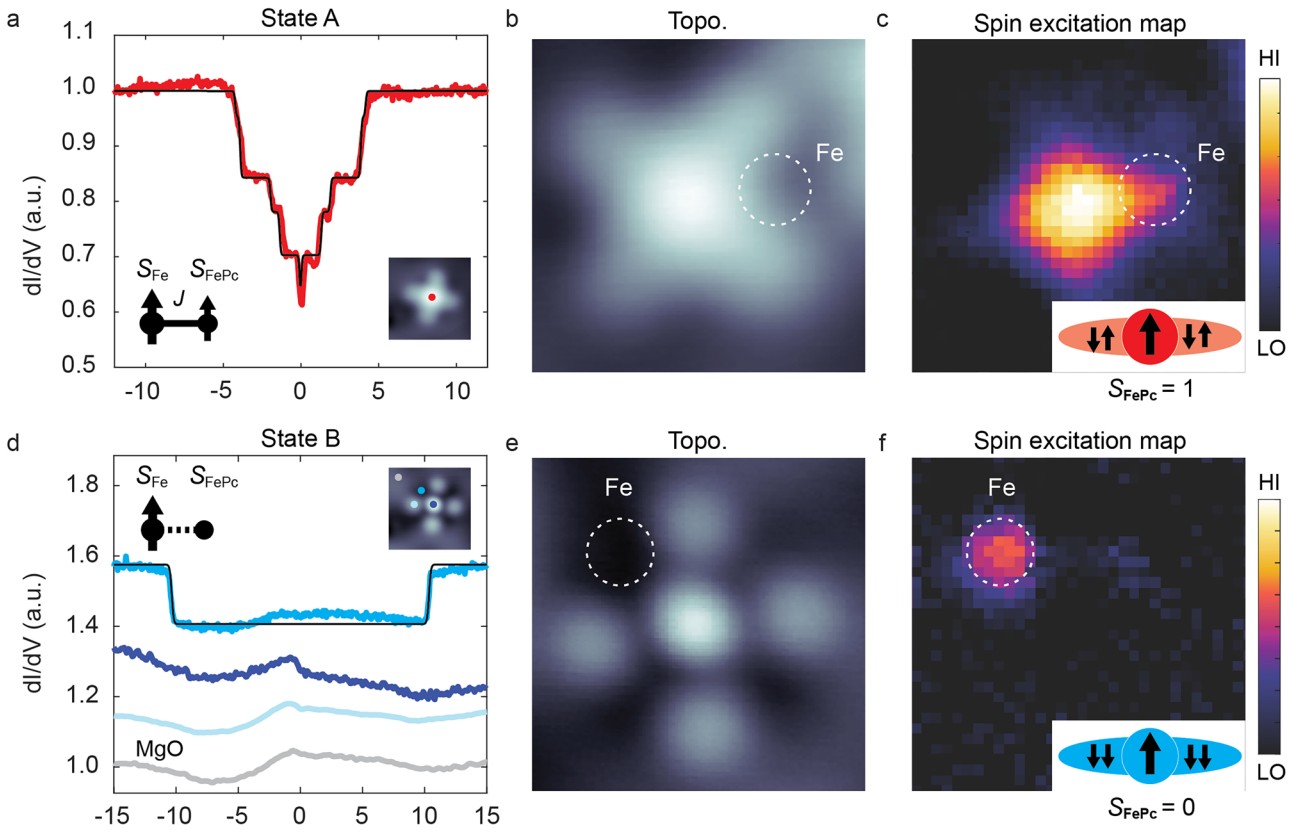

**Fig. 3 | Spin Structure of the Fe-FePc complexes. a** d$I$/d$V$ spectrum of State A acquired at the FePc center showing several steps originating from inelastic electron tunneling excitations [Step positions: ±(0.35, 1.30, 2.19, 3.90, 4.46) mV, setpoints: $V_{DC} = -30$ mV, $I = 100$ pA, $V_{mod} = 50$ μV]. The black curve is the fit using inelastic electron spin transport calculations[50], where an Fe spin $S = 2$ ($D = -0.86$ meV, $E = 0.11$ meV) is coupled to an FePc $S = 1$ ($D = 1.3$ meV), and the coupling strength is $J = -0.88$ meV. This is additionally illustrated in the sketch in the inset. **b** Topographic image of State A ($V_{DC} = -300$ mV, $I = 20$ pA), and **c** the corresponding pseudocolor spin excitation map. (setpoint: $V_{DC} = -30$ mV, $I = 60$ pA, $V_{mod} = 1$ mV). The spin signal is mapped by the difference [d$I$/d$V(V = 12$ mV) - d$I$/d$V(V = 0$ mV)]. The Fe adatom position is indicated by a dashed circle. Additional d$I$/d$V$ linecuts across the complex are shown in Supplementary Note 10. **d** d$I$/d$V$ spectra of State B measured at different positions [from top to bottom: Fe adatom,

FePc center, ligand and MgO (reference), indicated by colored dots in the inset ($V_{DC} = 30$ mV, $I = 80$ pA, $V_{mod} = 0.3$ mV]. The step at ±10.5 mV is fitted by using $S_{Fe} = 2$, $D = -3.5$ meV and $S_{FePc} = 0$ (black curve). **e** STM image of State B ($V_{DC} = 80$ mV, $I = 20$ pA) and **f** the corresponding spin excitation map of the complex (setpoint: $V_{DC} = 30$ mV, $I = 80$ pA, $V_{mod} = 0.8$ mV). The spin signal is mapped by the difference [d$I$/d$V(E = 15$ meV)−d$I$/d$V(E = 0$ meV)]. Insets in **c**, **f** depict the sketch of the FePc spin states in State A and B, based on DFT calculations (see also Supplementary Note 3): In State A, the ligand spin configuration is compensated (-0 μB), whereas in State B, their alignment results in a magnetic moment of -1.7 μB. In State B, the ligand spin is antiferromagnetically coupled to the magnetic moment of the central Fe atom (-1.8 μB), which results in a non-magnetic configuration of the total molecule spin moment (See Supplementary Note 3).

(Supplementary Note 9). We additionally resolve the excitation by a spin contrast map in Fig. 3b, c emphasizing that the spin contrast is mostly concentrated on the center of the molecule and extends to a certain degree towards the Fe adatom. The Fe adatom position is faintly visible as an enhanced IETS contrast between two of the lobes (indicated by a dashed circle). In contrast, the d$I$/d$V$ spectra of State B show no spin signatures at either the FePc center or the ligand site. This is further supported in measurements on the FePc site using spin-polarized tips (Supplementary Note 9). However, a step feature appears at -10.5 mV at the Fe adatom site, which is close to the inelastic excitation energy (-14 meV) of an isolated Fe adatom on MgO/Ag(001). The reduction in inelastic excitation energy likely arises from a different magnetic anisotropy. In the spin contrast map (Fig. 3f), this step feature is localized exclusively on the Fe adatom site (Fig. 3e). Thus, the measurements in Fig. 3 clearly demonstrate that the spin states are different between State A and B: State A shows a rich IETS pattern, suggesting that both spin centers host an unpaired spin and are likely coupled to each other. State B in contrast is spectroscopically silent on the FePc molecule, indicative of an $S = 0$ state. Only the Fe adatom shows inelastic excitations that are close to the features found for isolated Fe adatoms.

A similar change in the spin states is found in the DFT calculations (Fig. 1e, f; Supplementary Note 3). Table 1 summarizes the fractional charges found on the FePc center, the phthalocyanine ligand as well as the Fe adatom: According to the DFT calculations, the spin state of FePc in the complex transitions from a high-spin state ($S = 1$) in State A to a low-spin state ($S = 0$) in State B, while the Fe adatom retains a spin state of $S = 3/2$ in both configurations. In State A, FePc is antiferromagnetically coupled to the Fe adatom with an energy difference between ferromagnetic (FM) and antiferromagnetic (AFM) coupling ($E_{FM} - E_{AFM} = 4$ meV). In contrast, in State B, both the central Fe atom of FePc and the phthalocyanine ligand host a spin $S = 1$. Since both are preferentially antiferromagnetically coupled, they form a non-magnetic configuration, resulting in $S = 0$ configuration. Moreover, the central Fe atom of FePc and the Fe adatom exhibit a preference for ferromagnetic coupling ($E_{FM} - E_{AFM} = -45$meV). Thus, the change in spin configuration of FePc is rationalized by a change in the electronic occupation of both the central Fe spin and its ligand spin, while the Fe adatom maintains an unchanged spin state in both cases.

To gain a deeper understanding of the spin structure from the experiment, we simulate the d$I$/d$V$ spectra for both configurations

**Table 1 | Spin configuration in the two states found in experiment and DFT**

|  | FePc (Fe) | FePc (Pc) | FePc (total) | Fe adatom |
|---|---|---|---|---|
| State A (DFT) | $-2.09\,\mu_B$ $(S=1)$ | $0.01\,\mu_B$ $(S=0)$ | $-2.08\,\mu_B$ $(S=1)$ | $3.07\,\mu_B$ $(S=3/2)$ |
| State A (IETS) |  |  | $S=1$ | $S=2$ |
| State B (DFT) | $1.84\,\mu_B$ $(S=1)$ | $-1.69\,\mu_B$ $(S=1)$ | $0.16\,\mu_B$ $(S=0)$ | $2.91\,\mu_B$ $(S=3/2)$ |
| State B (IETS) |  |  | $S=0$ | $S=2$ |

The sign of the spin in DFT indicates the relative alignment between different spins. We find that the main difference between the two states is the additional occupation of the ligand spin in the case of State B.

using spin transport calculations[50]. For the complex in State A, both the correct position and intensity of IETS measurements can be best reproduced (black curves in Fig. 3a) using a Hamiltonian of the form

$$H = J\vec{S}_{FePc} \cdot \vec{S}_{Fe} + D_{FePc}S_{FePc,z}^2 + \left[D_{Fe}S_{Fe,z}^2 + E_{Fe}\left(S_{Fe,x}^2 - S_{Fe,y}^2\right)\right] \quad (1)$$

where $J = -0.88$ meV (FM) is the Heisenberg exchange coupling between the FePc and Fe adatom spins. In this simulation, FePc has a spin state of $S_{FePc} = 1$ with an out-of-plane magnetic anisotropy $D_{FePc} = 1.3$ meV. The Fe adatom has a spin state of $S_{Fe} = 2$ with an out-of-plane magnetic anisotropy $D_{Fe} = -0.86$ meV and an in-plane magnetic anisotropy $E_{Fe} = 0.11$ meV (See Supplementary Note 13 for the energy level diagram). For the complex in State B, the observed step feature at the Fe adatom is reproduced with $S_{Fe} = 2$ and $D_{Fe} = -3.5$ meV (Fig. 3d black curve) while the spin of FePc is set to $S_{FePc} = 0$, thus being non-magnetic. Although the spin state of the Fe adatom differs here from the DFT calculation, we suggest that it remains $S_{Fe} = 2$, since the IETS result closely resembles that of an isolated Fe adatom $(S_{Fe} = 2)$[36] and DFT also predicts $S = 2$ for an isolated Fe adatom (see solution H in Supplementary Note 3). The discrepancy in the spin state of Fe in proximity to FePc is attributed to the charge transfer from the adatom and its hybridization with both FePc and the surface. Also, spin models using $S_{Fe} = 3/2$ were tested but failed to reproduce the data for State A. In general, a great variety of other spin configurations were tested, of which the chosen set of spin states and parameters $J$, $D$ and $E$ provided the best agreement with the experimental spectra. While a tensorial exchange interaction $J$ could in principle offer a more realistic description in an anisotropic environment, we find that a simple scalar $J$ suffices to capture all experimentally observed features and, in addition, improves the interpretability. The use of an isotropic $J$ is consistent with prior studies of on-surface spin systems, including the exchange interaction between Fe atoms and FePc molecules[16,19].

In addition to the difference in Fe spin state, the spin coupling in State A is AFM in the DFT calculations, while experimentally we find FM coupling. We rationalize this by the fact that magnetic exchange couplings – being highly sensitive to interatomic distance and local coordination – are generally difficult to capture by DFT and that the coupling is overall rather weak for State A (DFT: $E_{FM} - E_{AFM} = 4$ meV; IETS: $J = -0.88$ meV). In general, we highlight that, due to the multiconfigurational character of FePc[51-53] and charge transfer with the MgO/Ag substrate[54] (Supplementary Note 3), accurately determining the electronic and magnetic configurations is challenging for DFT–as evidenced by the incorrect spin state of the Fe adatom. Nevertheless, our calculations qualitatively capture the adsorption of States A and B and elucidate the mechanism behind the bistable magnetic behaviour: Here, both DFT and IETS find a change from a paramagnetic (State A, $S_{FePc} = 1$) to a non-magnetic configuration (State B, $S_{FePc} = 0$) of the FePc molecule driven mainly by the change in spin occupation of the Pc ligand [inset of Fig. 3c, f]. Also, DFT calculations show the same change in spin state of pristine FePc without Fe adatom (Supplementary Note 3), emphasizing that the spin state change originates from the different adsorption site and orientation of the FePc (stabilized by the Fe in the complex).

In order to showcase the operation of this spin switch, we built a small structure that allows us to test its action on a target spin system.

The control mechanism is outlined in Fig. 4a while the structure is shown in Fig. 4b. It consists of a pristine FePc (Fig. 4b: top) acting as a simple $S = \frac{1}{2}$ target spin system with $g \approx 2$[19,20] as well as the bistable Fe-FePc switch (Fig. 4b: bottom) in direct proximity. The resonance frequency $f_0$ of the FePc is given by

$$hf_0 = g\mu_B B \quad (2)$$

Where $B$ is the external magnetic field, $h$ is Planck's constant and $\mu_B$ the Bohr magneton (for the discussion of residual tip field $B_{tip}$ see Supplementary Note 11). However, $f_0$ can be additionally shifted by neighbouring magnetically coupled spins[16,19,26]. This adds a term $hf_{dipolar} = \frac{\mu_0}{2\pi}\frac{1}{r^3}m_z^{FePc} \cdot m_z^{switch}$ to Eq. (2), that can shift the resonance frequency of the target spin as well, depending on whether the spin switch is $S = 0$ or $S = 1$. In a similar fashion, Heisenberg exchange interaction, $H = J \cdot S_{switch} \cdot S_{FePc}$, can shift $f_0$[20]. To test the influence of the spin switch, we measured an ESR spectrum[15,19] (see Methods) on the target FePc (Fig. 4c). With the switch in State B, we obtain a resonance frequency $f_0^B = 12.749$ GHz, which corresponds to good approximation to that of a spin ½ with $g \approx 2$. Subsequently, we applied a short voltage pulse to switch to State A (Fig. 4b: right). Now, using the same ESR parameters, we obtain $f_0^A = 12.802$ GHz as shown in Fig. 4c. In other words, the switching between State A and B of the close-by spin switch results in a frequency shift of $\Delta f_0 = 53$ MHz ($\approx 220$ neV) of the target FePc molecule. This shift is larger than expected for bare dipolar interaction (22 MHz), as estimated using the spin states from IETS results. We attribute the additional shift to ligand-mediated ferromagnetic exchange interaction between the two spins[19] (See Supplementary Note 12 for discussion on the spin-spin coupling).

To demonstrate a reversible switching between the two states, we established a switching routine in Fig. 4d. Here, we continuously monitor the ESR signal over time at the resonance condition of State A, i.e. $f_{probe} = f_0^A$. Consequently, a high ESR signal probes the ESR peak under resonance conditions (red range), while a low signal indicates that $f_0^A$ is shifted away from $f_{probe}$ (blue range). Next, we toggled the resonance condition between States A and B by a spin switch routine: This routine moved the tip from the target FePc to the spin switch and applied an STM voltage pulse for 3 s. After triggering the switch, the tip was immediately returned to the target FePc position for continued ESR readout. In total, we performed 12 switching attempts, 7 of which were successful (yellow triangles). The success rate here is not as high as that of an isolated spin switch (Figs. 2, 3), likely due to the presence of the nearby target FePc. The success rate could be further improved by fine tuning of the switching parameters (In particular for B→A, which accounts for 4 of the 5 failed attempts) and by simply applying multiple pulses at the same time. In addition, we found that only switching from State B to A actually required additional lateral movement of the tip: The switching from State A to B remained even feasible remotely, i.e. with the tip kept at the target FePc position.

In summary, we have experimentally demonstrated a simple atomic structure, consisting of an Fe adatom and an FePc molecule, that shows bistability in its ground state configuration. This bistable switch shows a distinct change in its magnetic structure. In particular, we were

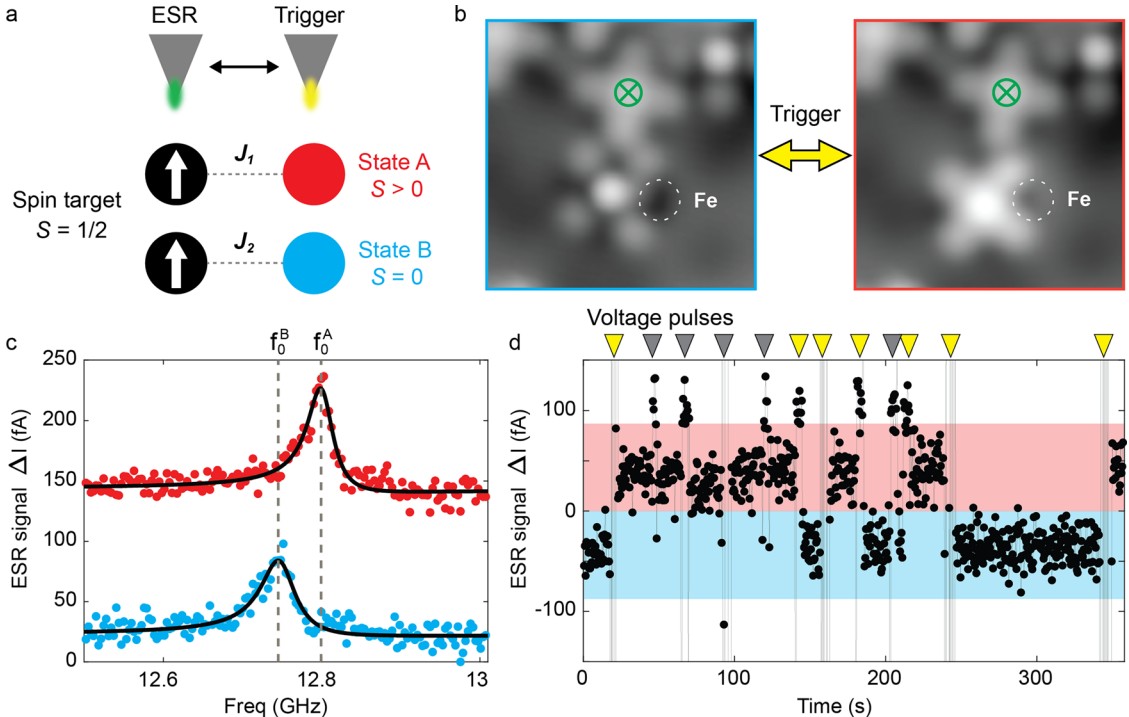

**Fig. 4 | Electron spin resonance (ESR) frequency shift using the spin switch.**
**a** Schematic of the experimental setup: an FePc spin ½ target (black ball with white arrow) is coupled to an Fe-FePc spin switch (red for State A, blue for State B). $J_1$ and $J_2$ denote the interaction strengths between the target spin and the spin switch in different states. The distance between the molecule centers can be found in Supplementary Note 14. The tip is used for ESR readout when positioned above the target spin. For triggering the switching events, it is moved above the spin switch. **b** STM images (3 nm × 3 nm; $V_{DC} = -100$ mV, $I = 20$ pA) showing an Fe-FePc spin switch (bottom) coupled to a target FePc molecule (top, green cross) at the edge of a molecular island. Switching events were induced by an STM voltage pulse (State B → A: $V_{DC} = -300$ mV, $I = 20$ pA; State A→ B: $V_{DC} = 300$ mV, $I = 500$ pA). **c** ESR signals recorded at the center of the FePc target spin, showing a 53 MHz shift between the resonance frequencies $f_0^A$ and $f_0^B$ of State A (top) and State B (bottom), respectively. (ESR conditions: $V_{DC} = 40$ m$V$, $I = 10$ pA, $V_{rf} = 8$ mV, $B_{ext} = 484$ mT). **d** ESR signal $\Delta I$

with fixed frequency set to $f_{probe} = f_0^A$ recorded over time. A high signal (red region) corresponds to on-resonance conditions of $f_0^A$, while a low signal (blue region) corresponds to off-resonance conditions of $f_0^A$. Triangles at the upper end of the graph indicate switching protocols, in which the tip is briefly moved to trigger the spin switch via corresponding voltage pulses (State B → A: $V_{DC} = -300$ mV, State A → B: $V_{DC} = 300$ mV) and return to the target spin as illustrated in a. In total 12 voltage pulses were attempted to induce the switching event, of which 7 were successful (yellow). Spikes in the current are due to the tip movement and change of the tunneling parameters for switching during the ESR measurement (ESR conditions: $V_{DC} = 40$ mV, $I = 10$ pA, $V_{rf} = 8$ mV, $B_{ext} = 508$ mT, $f_{probe} = 13.74$ GHz, tip travel speed = 18.4 nm/s. Note that the different frequency compared to (**c**) results from a different external magnetic field. All the voltage pulses and ESR measurements were done with closed feedback loop).

able to demonstrate its functionality by altering the resonance frequency of a target spin system. Our work prototypes the construction of a scalable switchable spin architecture in terms of single molecule machines. It highlights how the local energy landscape of on-surface objects such as atoms and molecules can be exploited by electric control of the STM tip to alter spin systems. The presented switch is applicable in on-surface spin qubit structures; however, we stress that the underlying operational principles may also be achieved through the chemical synthesis of specialized spin switch molecules: First, the addition of the Fe atom brings the two states closer in energy enabling the electric field-induced bistability. Thus, the Fe adatom and the surface play a crucial role here in promoting bistability, which instead could be realized through the incorporation of specific side groups in a single spin switch molecule. Second, the magnetic functionality of the switch is enabled by a change in the molecule's spin occupation loading a spin to its ligand, thereby compensating the total spin.

Thus, we envision the implementation of different classes of single molecule machines with the general ingredients presented here, that are essential to develop molecule-based spintronics and quantum information devices.

## Methods
The sample preparation was carried out in-situ at a base pressure of <5 × 10⁻¹⁰ mbar. The Ag(001) surface was prepared through several cycles

of Argon ion sputtering and annealing through e-beam heating. For MgO growth, the sample was heated up to 430 °C and exposed to a Mg flux for 20 minutes in an oxygen environment at 10⁻⁶ mbar leading to a MgO coverage of ~50% and layer thicknesses ranging from 2 to 5 monolayers. Subsequently, FePc was evaporated onto the sample held at room temperature using a home-built Knudsen cell at a pressure of 9 × 10⁻¹⁰ mbar for 90 seconds. Electron-beam evaporation of Fe was carried out for 21 seconds onto the cold sample. We determined the thickness of MgO layers through point-contact measurements on single Fe adatoms[55]. All experiments were carried out using a Unisoku USM1600 STM inside a homebuilt dilution refrigerator with a base temperature of 50 mK. An effective spin temperature of ~300 mK was estimated from ESR measurements of Fe dimers. Here, the intensities of the electron spin states depend on temperature[16], which we take as an estimate of the Boltzmann distribution in the experiment.

STM vertical manipulation was employed to build Fe-FePc complexes. Firstly, by positioning the tip above one of the ligands of the FePc molecule on MgO/Ag(001), the molecule is picked up by gently approaching the tip close to the molecule. Next, we apply a short STM voltage pulse at $V = 0.85$ V with an opened feedback loop. A sudden change of the tunneling signal can be observed, indicating a successful pick up of molecule. Then, a subsequent topography is recorded to ensure the pick-up. After that, the tip will be positioned above an Fe adatom on MgO/Ag(001) at a predefined position. Lastly, a similar

sequence of "pick up" is then applied to drop the molecule onto the Fe adatom.

Spin-polarized tips were prepared as follow: (1) individual Fe adatoms were transferred onto the Ag coated PtIr tip by STM vertical manipulation. (2) The spin polarization was then verified through the asymmetry in the differential conductance around zero bias in ($dI/dV$) measurements on FePc adsorbed on MgO/Ag(001). (3) Magnetic tips showing a high spin contrast were subsequently tested in the ESR-STM measurements (continuous wave). The radiofrequency (RF) voltage was applied on the tip-side of the junction using a RF generator (Rohde & Schwarz SMB100B). The RF voltage was combined with the DC tunnel bias using a Bias tee (Marki Microwave MDPX-0305). We used a digital lock-in amplifier (Stanford Research Systems SR860) to read out the ESR signal using an on/off modulation scheme at 323 Hz. Note that while the bias voltage was applied to the STM tip, all bias signs were inverted in the manuscript to follow the conventional definition of bias voltage with respect to the sample bias.

DFT calculations were performed using the VASP code[56]. The PBE form of the GGA exchange-correlation functional was used[57], and missing dispersion interactions in this functional were treated using the D3 scheme with Becke-Johnson damping[58]. The core electrons were treated by the projector augmented-wave method[59], and wavefunctions were expanded using a plane wave basis set with an energy cutoff of 400 eV. The Dudarev implementation of the LDA + U method[60] was used to treat the 3$d$ electrons of Fe, with $U_{eff} = U$-$J = 3$ eV, which has been used in previous FePc studies[51,61]. The MgO/Ag(001) surface was modeled using a slab formed by two MgO layers on top of four Ag layers, with a vacuum region of at least 15 Å, and a 6 × 6 surface unit cell. The position of all atoms in the unit cell except the two bottom Ag layers were relaxed until forces were smaller than 0.01 eV/Å. Corrections to potential and forces due to the presence of a dipole moment in the slab were applied[62]. Charge transfers and magnetic moments were determined by Bader analyses[63].

## Data availability
The data supporting the findings of this study are available in the article. Source data are provided with this paper.

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

## Acknowledgements

P.W. acknowledges funding from the Emmy Noether Programme of the DFG (WI5486/1-1), financing from the Baden Württemberg Foundation Program on Quantum Technologies (Project AModiQuS). P.W. and K.H.A.Y. acknowledge support and financing from the Centre for Integrated Quantum Science and Technology (IQST). P.G. and P.W. acknowledge financial support from the Hector Fellow Academy (Grant No. 700001123). R.R. and N.L. thank financial support from project PID2021-127917NB-I00 funded by MCIN/AEI/10.13039/501100011033, from project IT-1527-22 funded by the Basque Government, and from project ESiM no. 101046364 funded by the European Union.

## Author contributions

P.W. and W.H. conceived the research. W.H., K.H.A.Y., P.G., M.S., C.S., W.W. and P.W. set up the experiment and conducted the measurements. W.H., K.H.A.Y., P.G., M.S. and P.W. analyzed the experimental data. R.R. and N. L. performed the DFT calculations. W.H., K.H.A.Y., P.G., M.S., C.S., W.W. and P.W. discussed the results. W.H., K.H.A.Y. and P.W. wrote the manuscript with input from all authors. W.W. and P.W. supervised the project.

## Funding

## Competing interests

The authors declare no competing interests.
