## [Transparent Peer Review file · Nature Communications]

An electrically controlled single molecule spin switch

Corresponding Author: Dr Philip Willke

Version 0:

Reviewer comments:

Reviewer #1

(Remarks to the Author)

The report of Huang et al. describes a precise measurement of FePc (Pc-phthalocyanine) for its spin states. Using the bistability nature of the adsorbed molecule, the authors switch the molecule configuration which accompanying the magnetic property variation. The authors claim that this property can be used for future spin devices. The spin properties are revealed using inelastic tunneling spectroscopy of the spin excitation and ESR/STM method.

Though all topographic imaging and spectroscopy measurements, together with DFT calculations, were conducted with robust and professional techniques, the referee is afraid to say that the report has to stress the novelty of this work more in order to be published in Nature Communications. Both IETS spin excitation and ESR/STM were demonstrated for MPC molecule. A few groups can realize the latter technique, but it is not well explained why ESR/STM is indispensable for this work. The referee considers the manuscript requires major revision to highlight the novelty of this work.

Reviewer #2

(Remarks to the Author)

REPORT on the paper entitled:
" An electrically controlled single molecule spin switch "
by W. Huang et al

This paper convincingly shows that it is possible to construct a molecular spin switch, via STM tip-assisted on-surface assembly, that can be controlled electrically by the STM tip and is able to tune nearby magnetic states.

The switch consists of a molecular complex formed by a single Fe adatom and a FePc molecule adsorbed in close proximity on the surface of a 2 ML MgO/Ag(001) substrate, which is able to make transitions between two bistable states through the application of an STM bias voltage.

By dI/dV (IETS) spectroscopy the complex is shown to be magnetic in one configuration (state A) where both the FePc molecule and the Fe adatom possess a magnetic moment whereas in the other configuration (state B) only the Fe adatom is magnetic, the FePc molecule being magnetically silent. These findings are corroborated by fitting the dI/dV data by spin transport calculations based on a model spin Hamiltonian and DFT calculations for the two states, which provide also the value of the various magnetic moments.

Finally the spin switch complex is used to test its action on a target spin system.

To this purpose a small structure is built consisting of a pristine FePc molecule acting as a simple $S = 1/2$ target in direct proximity of a bistable Fe-FePc switch, so as to get a magnetic coupling between the two units..

With the tip on the target, the change in Larmor frequency of the target is measured by the technique of Electron Spin Resonance (ESR) while the nearby bistable complex is switched from state A to state B by a change of bias voltage, signaling the magnetic functionality of the switch.

I think this work is a notable achievement that will have an impact on spin-based quantum

technologies. State-of-the-art methodology is used to achieve the result, the quality of the data is very good for a general reader like me and enough details are given, both in the main text and in the supplementary material, for the work to be reproduced, except perhaps for the DFT calculations, as discussed below.

Also the presentation, that follows the logical sequence described above, is functional to a good understanding of the intention of the authors to convey their point of view.

However there are few points in the paper that deserve some comments and/or clarifications.

a) It is known that FePc has been shown experimentally to be a mostly isotropic spin $S = 1/2$ system when adsorbed on MgO/Ag(001). Since the spin of the FePc molecule in the gas phase is $S = 1$, the interaction of the molecule with the substrate has the effect of reducing the spin value.

However in the complex of type A, i.e. a pristine FePc in close proximity with an Iron adatom, the DFT calculations presented by the authors shows that the spin goes back to the original value $S = 1$ (Table 1). In order to show an internal consistency of the theoretical framework it would be nice to present a similar calculation (perhaps in the Supplementary Material) for pristine FePc molecule alone.

This request comes from the fact that the DFT approach, even if implemented in the framework of the LDA + U method, is unable to capture the subtleties of the molecular multiplet structure of FePc, which shows ~ 0.1 eV separation between the ground and first excited state, not to speak of the spin-orbit interaction with half the strength (0.05 eV) (Natoli, Phys.Rev.B 97, 155139 (2018) and Phys.Rev.B 98, 195108 (2018)). Certainly the DFT results can be used as a guidance to follow the variation of physical parameters in similar situations, but one should be sure of some internal consistency of the framework used. The authors themselves notice a discrepancy between the spin value found for the Fe adatom ($S=3/2$) in contrast with their finding of an $S=2$ (their IETS results closely resemble that of an isolated Fe adatom with $S=2$). Also the magnetic coupling between the FePc molecule and the Fe adatom in the complex is found to be antiferromagnetic, at variance with their finding of an Heisenberg exchange J with negative value (ferromagnetic coupling). All this is not surprising, since BCC ion is a ferromagnet and FCC is antiferromagnetic, with the exchange interaction strongly dependent on the interatomic distance.

b) Fig. 2d compares the absolute ground state energies of states A and B with and without Fe adatom from DFT calculations. It is clear from Fig.s 1e and 1f what is the position of the two complexes with Fe adatom with respect to the underlying MgO lattice (see Fig.s 1c and 1d). So one can start from these two positions and minimize the total DFT energy by relaxing the position of all the atoms of the cluster taken into account (see Method Section). The difference in energy between the two configurations (51 meV) is a dependable figure since DFT is used within its original theoretical framework. It is not clear the procedure followed to establish the same quantity for the states without Fe adatom. I can imagine that the initial position of state A without Fe adatom coincides with the position of a pristine FePc. What about the initial position of a B state and how it is defined? Does one take the same position for the B state with Fe adatom, omit this latter, and then minimize again the total energy? I do not expect that two isolated pristine FePc molecules sit on the surface in two different configurations, but one never knows, due to the many uncontrollable sorts of interaction between the molecule and the substrate that can engender metastable states.

I ask this because I would like to understand

better the origin and the meaning of the energy difference (296 meV) in this case.

Obviously the relevant physical quantity here is the figure of 51 meV, which explains why its is possible to switch from state A to state B in presence of an Fe adatom with a modest variation of bias voltage.

c) Correctly, the authors use the DFT calculations to establish the magnetic properties (spin state) of the various ground states of the complexes under investigation.

To interpret their IETS measurements, which require excited states, and to model the operation of the spin switch, they use a model spin Hamiltonian based on the spin values suggested by the DFT calculations, like in their Eq. (1), with an added Zeeman term.

This approach is customary in the literature for this type of experiments, with various degrees of success. In this particular work the agreement between the predictions of the model and the experimental observations presents some problems, which however do not invalidate the main conclusions concerning the magnetic functionality of the switch. They all signal that the structure of the excited states of the systems studied is not fully captured by the model used. For example, although the shift in the Larmor

frequency of the test spin ($S=1/2$) is unquestionable and proves the magnetic functionality of the switch, there are problems both with the value of the expected shift, its magnitude and direction and the fact that only one magnetic transition is observed, honestly recognized and discussed by the authors. Even the observed values for the Larmor frequencies f_0^A and f_0^B around 12.8 GHz signal that there is a problem with the simple $S=1/2$ spin model of the test FePc molecule, since $g\mu_B/h$ corresponds to 28 GHz/T and for the value of the external magnetic field used in the experiment (484 mT) one obtains 13.55 GHz. Probably a different approach (Natoli, loc. cit.) could be useful, but I recognize that this is outside the scope of the present paper.

In my opinion, this is a very good paper.

(Calogero R. Natoli).

PS: delete lines from 350 to 364, since they are a repetition of lines from 334 to 348.

Reviewer #3

(Remarks to the Author)
NCOMMS-25-22844 By Wantong Huang et al

In this manuscript, the authors use a system consisted of a Fe adatom coupled to an iron phthalocyanine (FePc) molecule, and their response to the tip-voltage of a scanning tunneling microscope (STM). Using inelastic electron tunneling spectroscopy (IETS) they provide data on changes in configuration upon tip-voltage trigger. Accordingly they attribute this to a change between a paramagnetic and a non-magnetic spin configuration. In addition, they use in-situ Electron Spin Resonance-frequency shifts to probe the response of the target FePc spin within a spin-spin coupled structure. This is an interesting research providing high resolution IETS, STEM data and ESR-f data. However in its current form it suffers from data-interpretation inconsistencies and modeling-inadequacy. Given that the studied system FePc+Fe atom on MgO/Ag(001) is a well studied one, including STEM, thus the present work lacks the cutting-edge novelty and rigor for publication in Nature Communications

RECOMMENDATION Reject

Specific comments

As stated by the authors the FePc on MgO/Ag(001) is a isotropic spin $S = 1/2$ system when adsorbed while individual Fe adatoms are spin $S = 2$. In the STEM-set-up the authors suggest that the experimentally observed states are State A (IETS) $S_{FePc} = 1$ and in State B $S_{FePc} = 0$ while S_{Fe} adatom is (IETS) $= 2$. This indicates that the tip-voltage exerts significant influence on the electron-spin configuration of the system. However the authors do not clarify this, as in due.

In the whole body of the analysis the authors do not take into account the role of orbital-angular momentum, L . This would be the means to incorporate the effect of the tip-voltage to the spin-changes of the FePc from $S=1/2$ (with zero tip-voltage), to $S=1$ or 0 (under non-zero tip-voltage plus the coupling to the Fe adatom).

The authors offer several options to discuss inconsistencies between their data and the theoretical predictions e.g. the DFT and IETS and the Resonance frequency shift of FePc target induced by spin switch. I suggest the authors to take into account the role of L –including an electric/spin-Hamiltonian term– and try to clean-up which of these inconsistencies can be fixed.

The authors use a simple spin-Hamiltonian

$$H = J S_{FePc} \cdot S_{Fe} + [D_{FePc} S_{FePc}^2 + E_{Fe} (S_{Fe,x}^2 - S_{Fe,y}^2)]$$

to describe their system. This H entails a simple scalar J between S_{FePc} and S_{Fe} i.e. instead of a J -tensor, as should be in a more complete approach. The authors should justify this approach. This is critical i.e. the FePc/Fe-atom system on the MgO/Ag(001) plane plus the tip-electric field is a highly-anisotropic system that –in general– is expected to be represented by a non-isotropic- J .

The DFT calculations are inconclusive in present work. As recognized by the authors in their text DFR failed to predict the Spin-states (if we consider that the experimental IETS are correct). I suggest the authors to remove the DFT part from the manuscript and its –perplexing– discussion with the experimental data. In addition, it is not clear in the description of the DFT-method how the tip-electric-field effect has been taken into account.

Version 1:

Reviewer comments:

Reviewer #1

(Remarks to the Author)

The authors have well revised the introduction to emphasize the novelty of this report. The referee considers that the manuscript should be published as it is.

Reviewer #2

(Remarks to the Author)

REPORT on the revised version of the paper entitled:
" An electrically controlled single molecule spin switch "
by W. Huang et al

I am completely satisfied with the revised version of the paper with its annex of supplemental material (SM). The authors answered exhaustively to my comments and questions, clarifying also the origin of the measured Larmor frequency in the target ($S = 1/2$) spin system (mistakenly I had interpreted B_{ext} as the total applied field including the residual tip magnetic field).

In particular I appreciated the new section 3.2 of the SM where the authors discuss the electronic and spin states of the various systems encountered in their work (pristine FePc and complexes A and B) based on DFT calculations. They point out that the main physical effect in these systems is the charge transfer from the substrate to the adsorbed system which is well described by the DFT approach. There is however a further effect not correctly taken into account by this method, which is the description of the multi-configuration interaction of the various excited states of the complex so obtained, which at the end determines its spin state and the correct final amount of the charge transfer. However the authors are aware of this limitation of DFT and "cum grano salis" illustrate the evolution of the physical properties of the various complexes in a convincing way.

I think this paper deserves to be brought to the attention of the general scientific community.

(Calogero R. Natoli)

Reviewer #3

(Remarks to the Author)

In their Revised manuscript the authors made some text-improvements and provide opinions on my original comments. Although this is a fair response by the esteemed colleague, I consider that this revised-manuscript lacks the undisputable coherence and cutting-edge conclusiveness, than would qualify it for publication in Nature Communications Journal.

From their replies and the Reviewers comments it is obvious that there are issues and caveats in ESR data analysis (spin states, anisotropy-tensors), as well as in the assumptions needed to be done to accept the inadequacies of DFT to provide all key-parameters.

I suggest Decline from publication in Nature Communications Journal.

REVIEWER COMMENTS

Reviewer #1 (Remarks to the Author):

The report of Huang et al. describes a precise measurement of FePc (Pc-phthalocyanine) for its spin states. Using the bistability nature of the adsorbed molecule, the authors switch the molecule configuration which accompanying the magnetic property variation. The authors claim that this property can be used for future spin devices. The spin properties are revealed using inelastic tunneling spectroscopy of the spin excitation and ESR/STM method.

Though all topographic imaging and spectroscopy measurements, together with DFT calculations, were conducted with robust and professional techniques, the referee is afraid to say that the report has to stress the novelty of this work more in order to be published in Nature Communications. Both IETS spin excitation and ESR/STM were demonstrated for MPc molecule. A few groups can realize the latter technique, but it is not well explained why ESR/STM is indispensable for this work. The referee considers the manuscript requires major revision to highlight the novelty of this work.

Reply: We thank the reviewer for the comments. We agree that the novelty could have been emphasized more clearly and we have revised the introduction accordingly. In short, ESR-STM is essential in our study for three primary reasons: ESR-STM uniquely provides MHz (neV) energy resolution, which is critical for resolving the subtle magnetic interactions between the bistable spin switch and a nearby spin. In contrast, conventional inelastic tunneling spectroscopy (IETS) can only reach μeV to meV resolution in STM experiments. In other words, the observed 50 MHz shifting (Fig. 4) would not be possible to detect by IETS. Second, spin resonance is the method of choice for controlling and reading out spin qubits. Thus, its use is important to showcase that a quantum device architecture could work based on a molecular scale. Third, ESR provides a direct proof of the change in magnetic field of the switch. For instance, often the change in magnetic properties for on-surface molecules is inferred from the appearance or disappearance of a Kondo resonance (e.g., Ref. Nature Communications 3, 938 (2012)). While this is a valid approach, the Kondo coupling also drastically alters the magnetic properties of the switch and consequently renders it useless as a spin switch.

As the reviewer has pointed out, previous studies have demonstrated ESR-STM on MPc molecules (e.g., FePc, TbPc₂), and, separately, magnetic molecular switches have been

realized before (e.g., Ref. Gruber et al. Nature Nanotechnology 15, 18–21 (2020)). The novelty of our work lies in integrating these functionalities into a single system: We demonstrate, for the first time, a bistable molecular spin switch with electrical control and showcase its functionality in a proof-of-concept device. This combination of molecular electronics, spintronics, and quantum control has not been realized previously, and we highlight this aspect along with why ESR-STM is indispensable now more clearly at the end of the revised introduction:

"...Combined inelastic electron tunnelling spectroscopy (IETS) measurements and DFT calculations, reveal a reversible change in the molecular spin states, switching between configurations, $S > 0$ and $S = 0$. We demonstrate the functionality of the spin switch by tuning the resonance frequency of a nearby FePc target spin center, that is magnetically coupled to a switch: Here, we utilize ESR-STM and its high energy resolution (\sim neV) to i) directly detect the change in the magnetic dipole field of the switch and ii) resolve the weak intermolecular magnetic coupling. As ESR is compatible with coherent spin control²⁰, our system serves as a proof-of-concept device for reversible, switchable qubit-qubit interactions within a molecule-based quantum platform. Bridging the fields of molecular machines, local bottom-up assembly as well as spin-based quantum control, this work provides a foundational step towards a scalable molecular quantum architecture.

"

Reviewer #2 (Remarks to the Author):

REPORT on the paper entitled:

" An electrically controlled single molecule spin switch " by W. Huang et al

This paper convincingly shows that it is possible to construct a molecular spin switch, via STM tip-assisted on-surface assembly, that can be controlled electrically by the STM tip and is able to tune nearby magnetic states. The switch consists of a molecular complex formed by a single Fe adatom and a FePc molecule adsorbed in close proximity on the surface of a 2 ML MgO/Ag(001) substrate, which is able to make transitions between two bistable states through the application of an STM bias voltage. By dI/dV (IETS) spectroscopy the complex is shown to be magnetic in one configuration (state A) where both the FePc molecule and the Fe adatom possess a magnetic moment whereas in the other configuration (state B) only the Fe adatom is magnetic, the FePc molecule being

magnetically silent. These findings are corroborated by fitting the dI/dV data by spin transport calculations based on a model spin Hamiltonian and DFT calculations for the two states, which provide also the value of the various magnetic moments. Finally the spin switch complex is used to test its action on a target spin system. To this purpose a small structure is built consisting of a pristine FePc molecule acting as a simple $S = 1/2$ target in direct proximity of a bistable Fe-FePc switch, so as to get a magnetic coupling between the two units. With the tip on the target, the change in Larmor frequency of the target is measured by the technique of Electron Spin Resonance (ESR) while the nearby bistable complex is switched from state A to state B by a change of bias voltage, signaling the magnetic functionality of the switch.

I think this work is a notable achievement that will have an impact on spin-based quantum technologies. State-of-the-art methodology is used to achieve the result, the quality of the data is very good for a general reader like me and enough details are given, both in the main text and in the supplementary material, for the work to be reproduced, except perhaps for the DFT calculations, as discussed below.

Also the presentation, that follows the logical sequence described above, is functional to a good understanding of the intention of the authors to convey their point of view.

Reply: We thank the reviewer for the support and appreciate the insightful and constructive comments, which we believe have improved the manuscript. We address them below.

However there are few points in the paper that deserve some comments and/or clarifications.

a) It is known that FePc has been shown experimentally to be a mostly isotropic spin $S = 1/2$ system when adsorbed on MgO/Ag(001). Since the spin of the FePc molecule in the gas phase is $S = 1$, the interaction of the molecule with the substrate has the effect of reducing the spin value. However in the complex of type A, i.e. a pristine FePc in close proximity with an Iron adatom, the DFT calculations presented by the authors shows that the spin goes back to the original value $S = 1$ (Table 1). In order to show an internal consistency of the theoretical framework it would be nice to present a similar calculation (perhaps in the Supplementary Material) for pristine FePc molecule alone. This request comes from the fact that the DFT approach, even if implemented in the framework of the

LDA + U method, is unable to capture the subtleties of the molecular multiplet structure of FePc, which shows ~ 0.1 eV separation between the ground and first excited state, not to speak of the spin-orbit interaction with half the strength (0.05 eV) (Natoli, Phys.Rev.B 97, 155139 (2018) and Phys.Rev.B 98, 195108 (2018)). Certainly the DFT results can be used as a guidance to follow the variation of physical parameters in similar situations, but one should be sure of some internal consistency of the framework used. The authors themselves notice a discrepancy between the spin value found for the Fe adatom ($S=3/2$) in contrast with their finding of an $S=2$ (their IETS results closely resemble that of an isolated Fe adatom with $S=2$). Also the magnetic coupling between the FePc molecule and the Fe adatom in the complex is found to be antiferromagnetic, at variance with their finding of an Heisenberg exchange J with negative value (ferromagnetic coupling). All this is not surprising, since BCC ion is a ferromagnet and FCC is antiferromagnetic, with the exchange interaction strongly dependent on the interatomic distance.

Reply: We thank the reviewer for the helpful suggestion. We have revised the text and included additional supplemental sections in order to address the points. We answer the different remarks by the reviewer separately below.

1. DFT Calculations of FePc with and without a closeby Fe adatom

The referee is absolutely right in pointing out the problems of DFT in fully describing the complex multiconfigurational nature of FePc. Indeed, the main purpose of our DFT calculations in this work is to identify the bistable states observed in the experiment.

Following the suggestions of the referee, we have added to the Supplementary Information an extra figure [Supplementary Fig. 4] with the electronic structure of states A and B with and without an extra adatom, and we have completed Supplementary Table 1. For clarity, we also pasted it here.

Supplementary Fig. 4. DFT calculations of spin densities. Top view and side view images showing the spin densities of States A and B (a) with Fe adatom and (b) without an extra Fe adatom. Blue (red) color represents majority (minority) spin.

We highlight a) that even without the Fe atom, we observe a change in spin state, which emphasizes that the latter originates from the different adsorption site and orientation of the FePc [stabilized by the Fe, see question b) of the reviewer below]. b) These results however are in contrast with the spin state observed experimentally for isolated FePc, which shows a $S=1/2$ [Nat. Chem. 14, 59–65 (2022)]. In the new supplementary section 3.2 we argue that a potential reason for that deviation is the system's sensitivity to charge transfer, which in turn is strongly dependent on the experimental conditions.

The new supplementary section 3.2 reads:

“3.2. Spin Configuration of isolated FePc and complex A and B

In this section we discuss in greater detail the qualitative picture of the FePc spin states based on the DFT calculations and additionally comment on the spin state of the isolated FePc. Accurately modeling this system is challenging - not only due to the intrinsic complexity of FePc itself stemming from the multiorbital ground state¹⁻³ - but also because of its nontrivial

interaction with the MgO/Ag(100) substrate: There is a significant charge transfer from the surface to the molecule, with FePc gaining approximately 1.5-2.0 electrons (see $\Delta N[\text{FePc}]$ in Supplementary Table 1 and Supplementary Fig. 5). This charge is transferred to the degenerate LUMO, which is mainly localized on the phthalocyanine (Pc) ligand. The electronic and magnetic structure of FePc is then dictated by how this additional charge is distributed. In State A, the charge rearrangement results in a nearly vanishing net magnetic moment on the Pc ligand (see $\mu[\text{Pc}]$ in Supplementary Table 1 and Supplementary Fig. 5c), yielding an overall spin state of approximately $S \approx 1$, primarily localized on the Fe center ($\sim 2.0 \mu_B$). In contrast, in State B, the excess charge induces a Pc-centered magnetic moment of $(1.6 - 1.8) \mu_B$, antiferromagnetically coupled to the Fe spin, leading to a total spin state of $S \approx 0$. This qualitative picture holds whether or not a nearby Fe adatom is present (Supplementary Fig. 4). Thus, the overall spin state of FePc is primarily governed by the magnitude and organization of the surface-induced charge transfer.

We note, that the spin state results of DFT for FePc (Supplementary Fig. 5b) without a nearby Fe adatom differ from those observed experimentally for isolated FePc, which exhibits a spin state of $S = 1/2$ - not $S = 1$ as in the case of State A. We attribute this discrepancy between DFT and experiment to the system's sensitivity to charge transfer, which in turn is strongly dependent on the experimental conditions. In DFT, excess charge transfer from the substrate is overestimated and goes to the ligand (Supplementary Fig. 5b), leading to a compensated ligand spin and an incorrect overall spin state for pristine FePc. Reducing the number of electrons on the ligand restores an uncompensated spin and yields the correct result. For example, combined experimental and theoretical studies have shown that the presence of interstitial oxygen between the MgO layers and Ag(001) significantly alters the charge transfer to the molecule by modifying the surface work function⁵. Therefore, in order to accurately describe pristine FePc, one not only has to correctly describe the molecule itself, but it is necessary to simultaneously account for the interaction with the surface, including the charge transfer. Nevertheless, the DFT calculations presented here effectively capture the qualitative electronic configurations associated with States A and B in the complex, and how these lead to two bistable magnetic states for FePc."

Additionally, we also comment on the magnetic moment of Fe adatoms in Supplementary Section 3.3:

"For the magnetic moment of the Fe adatom, we obtain $S = 3/2$ in both states A and B in the complex. However, for an isolated Fe adatom using the same computational parameters, we obtain $S = 2$ ($\mu = 3.90 \mu_B$), in agreement with the experimental observations⁶. When the Fe adatom is sufficiently far away (as in the solution H in Supplementary Fig. 3 and Table 1), a

spin state of $S=2$ is obtained. Therefore, the change in the spin state is a result of the charge transfer of the adatom and its hybridization with FePc and the surface.”

2. Sign of the Exchange Coupling

The reviewer is correct, that our DFT results show antiferromagnetic exchange ($J > 0$), whereas the IETS and ESR-STM experiments are consistent with a **ferromagnetic** Heisenberg exchange ($J < 0$). We agree that such discrepancies are not unexpected, as the sign and magnitude of the exchange interaction are highly sensitive to atomic distances and local coordination. We highlight this in the discussion of the results now:

‘We rationalize this by the fact that magnetic exchange couplings – being highly sensitive to interatomic distance and local coordination – are generally difficult to capture by DFT and that the coupling is overall rather weak for State A.’

b) Fig. 2d compares the absolute ground state energies of states A and B with and without Fe adatom from DFT calculations. It is clear from Figs 1e and 1f what is the position of the two complexes with Fe adatom with respect to the underlying MgO lattice (see Figs 1c and 1d). So one can start from these two positions and minimize the total DFT energy by relaxing the position of all the atoms of the cluster taken into account (see Method Section). The difference in energy between the two configurations (51 meV) is a dependable figure since DFT is used within its original theoretical framework.

It is not clear the procedure followed to establish the same quantity for the states without Fe adatom. I can imagine that the initial position of state A without Fe adatom coincides with the position of a pristine FePc. What about the initial position of a B state and how it is defined? Does one take the same position for the B state with Fe adatom, omit this latter, and then minimize again the total energy? I do not expect that two isolated pristine FePc molecules sit on the surface in two different configurations, but one never knows, due to the many uncontrollable sorts of interaction between the molecule and the substrate that can engender metastable states. I ask this because I would like to understand better the origin and the meaning of the energy difference (296 meV) in this case. Obviously the relevant physical quantity here is the figure of 51 meV, which explains why its is possible to switch from state A to state B in presence of an Fe adatom with a modest variation of bias voltage.

Reply: We agree with the reviewer that we should elaborate in detail on our methodology. We do this now in the Supplementary Section 3.1 (see below).

We here briefly explain the procedure followed to search for states A and B. Following our lattice analysis of the experimental results (Supplementary Fig. 2), we explored different

configurations: some of them had the Fe atom of FePc on a top position, while in others it was in a bridge position (with respect to O atoms in MgO, see Supplementary Fig. 3). The summary of all differences in energy, charge transfer ΔN as well as magnetic moments are given in Supplementary Table 1. States A and B were identified as the lowest energy ones on top and bridge positions, respectively (Supplementary Fig. 4a). The configurations with and without an extra Fe adatom show the same electronic configuration (now presented in the new Supplementary Fig. 4). We thank the referee for the suggestion of starting from state B geometry, removing the Fe adatom, and fully relaxing the system. After doing so, we have arrived to a new candidate for state B *without* an adatom with a lower energy difference of 187 meV. The electronic configuration remains identical. Although lower than the previous value of 296 meV, the new value is still significantly higher than the value in presence of an Fe adatom (51 meV). Therefore, the conclusions of our study remain unaltered, showing the stabilization of state B by the presence of Fe adatoms. We updated the figures and tables accordingly.

Finally, let us note that for state A we already used a procedure similar to the one suggested by the referee to search for configurations with an Fe adatom. We started from the most stable configuration of FePc on a top position and added an Fe adatom at different positions in order to look for different configurations. We have now updated them in supplementary section 3.1:

“3.1. Details on the energies of different FePc and complex configurations.

To rationalize the bistability and highlight the stabilizing role of the Fe atom on the complex, we systematically compared several configurations of FePc, both with and without an Fe adatom. Following our lattice analysis of the experimental results (Supplementary Fig. 2), we explored different configurations: some of them had the Fe atom of FePc on a top position, while in others it was in a bridge position (with respect to O atoms in MgO, see Supplementary Fig. 3). The summary of all differences in energy, charge transfer ΔN as well as magnetic moments are given in Supplementary Table 1. States A and B were identified as the lowest energy ones on top and bridge positions, respectively (Supplementary Fig. 4a). Here, the energy difference is 51 meV. In order to compare these results to pristine FePc in State A and State B (without Fe adatom) as done in Fig. 2d in the main text, we began from State A and State B geometry with Fe adatom (Supplementary Fig. 4a), then removed the Fe adatom, and fully relaxed the system. The resulting energy difference between pristine FePc in State A and State B is 187 meV. The electronic configuration remains mostly identical (Supplementary Table 1). This comparison highlights the stabilizing role of the Fe adatom.

”

c) Correctly, the authors use the DFT calculations to establish the magnetic properties (spin state) of the various ground states of the complexes under investigation. To interpret their IETS measurements, which require excited states, and to model the operation of the spin switch, they use a model spin Hamiltonian based on the spin values suggested by the DFT calculations, like in their Eq. (1), with an added Zeeman term. This approach is customary in the literature for this type of experiments, with various degrees of success. In this particular work the agreement between the predictions of the model and the experimental observations presents some problems, which however do not invalidate the main conclusions concerning the magnetic functionality of the switch.

They all signal that the structure of the excited states of the systems studied is not fully captured by the model used. For example, although the shift in the Larmor frequency of the test spin ($S=1/2$) is unquestionable and proves the magnetic functionality of the switch, there are problems both with the value of the expected shift, its magnitude and direction and the fact that only one magnetic transition is observed, honestly recognized and discussed by the authors.

Even the observed values for the Larmor frequencies f_0^A and f_0^B around 12.8 GHz signal that there is a problem with the simple $S=1/2$ spin model of the test FePc molecule, since $g\mu_B/h$ corresponds to 28 GHz/T and for the value of the external magnetic field used in the experiment (484 mT) one obtains 13.55 GHz. Probably a different approach (Natoli, loc. cit.) could be useful, but I recognize that this is outside the scope of the present paper.

Reply: We apologize for the confusion regarding the resonance frequency. The observed deviation between the expected resonance frequency (13.55 GHz at 484 mT) and the measured value (~ 12.8 GHz) arises from the additional magnetic field contribution of the spin-polarized magnetic tip, B_{tip} : B_{tip} shifts the ESR transitions of on-surface spin systems in ESR-STM experiments (See e.g. Nature Physics volume 15, pages1005–1010 (2019)) and depends on the exact location and distance between tip and spin center. This tip field, which is here antiferromagnetically coupled to the molecular spin, effectively reduces the total magnetic field. The resonance frequency is given by $hf_0 = g\mu_B(B_{ext} + B_{tip})$, and in our case, the tip field accounts for a downward shift of approximately -0.75 GHz. We had omitted this detail in the initial manuscript for readability, but we agree that it is important factor and will clarify this in the revised version.

In my opinion, this is a very good paper.

(Calogero R. Natoli).

PS: delete lines from 350 to 364, since they are a repetition of lines from 334 to 348.
Reply: Thanks for the suggestion. We have revised that.

Reviewer #3 (Remarks to the Author):

NCOMMS-25-22844 By Wantong Huang et all

In this manuscript, the authors use a system consisted of a Fe adatom coupled to an iron phthalocyanine (FePc) molecule, and their response to the tip-voltage of a scanning tunneling microscope (STM). Using inelastic electron tunneling spectroscopy (IETS) they provide data on changes in configuration upon tip-voltage trigger. Accordingly they attribute this to a change between a paramagnetic and a non-magnetic spin configuration. In addition, they use in-situ Electron Spin Resonance-frequency shifts to probe the response of the target FePc spin within a spin-spin coupled structure. This is an interesting research providing high resolution IETS, STEM data and ESR-f data. However in its current form it suffers from data-interpretation inconsistencies and modeling-inadequacy. Given that the studied system FePc+Fe atom on MgO/Ag(001) is a well studied one, including STEM, thus the present work lacks the cutting-edge novelty and rigor for publication in Nature Communications

Reply: We thank the reviewer for the comments which helped to improve and to strengthen our message. To avoid any confusion, we would like to clarify in the beginning that our work does not involve scanning transmission electron microscopy (STEM). The experimental techniques employed are (STM), inelastic electron tunneling spectroscopy (IETS), and electron spin resonance scanning tunneling microscopy (ESR-STM).

Regarding the comment on data-interpretation, we have revised the manuscript to improve clarity and address and comment on inconsistencies (see below). The reviewer is correct that Fe and FePc *separately* are well studied systems on MgO/Ag(100), but the key novelty is a combination of the two into a coupled spin system: This allows to demonstrate the electric control of a bistable molecular spin switch and its ability to modify the resonance frequency of the nearby spin. We agree that this novelty has not been highlighted clearly enough (see reply to reviewer No. 1) and we have now revised the introductory paragraph accordingly. To restate the answer to Reviewer 1:

“The novelty of our work lies in integrating these functionalities into a single system: We demonstrate, for the first time, a bistable molecular spin switch with electrical control **and**

showcase its functionality in a proof-of-concept device. This combination of molecular electronics, spintronics, and quantum control has not been realized previously, and we highlight this aspect now more clearly in the revised introduction.”

RECOMMENDATION Reject

Specific comments

As stated by the authors the FePc on MgO/Ag(001) is a isotropic spin $S = 1/2$ system when adsorbed while individual Fe adatoms are spin $S = 2$. In the STEM-set-up the authors suggest that the experimentally observed states are State A (IETS) $S_{\text{FePc}} = 1$ and in State B $S_{\text{FePc}} = 0$ while $S_{\text{Fe adatom}} = 2$. This indicates that the tip-voltage exerts significant influence on the electron-spin configuration of the system. However the authors do not clarify this, as in due.

In the whole body of the analysis the authors do not take into account the role of orbital-angular momentum, L . This would be the means to incorporate the effect of the tip-voltage to the spin-changes of the FePc from $S=1/2$ (with zero tip-voltage), to $S=1$ or 0 (under non-zero tip-voltage plus the coupling to the Fe adatom).

Reply: We thank the reviewer for raising this point. However, we would like to clarify that there appears to be a conceptual misunderstanding regarding the role of the tip voltage in our experiment: The voltage pulse via the STM tip is not a continuous field that alters the spin configuration; rather, it is only applied shortly above a threshold to induce a switching event in the molecule spin configuration. Importantly, the tip voltage is not maintained during the IETS or ESR measurements (only a small voltage that does not induce any bistability). Consequently, the observed spin (e.g., $S=1$, or $S=0$ for FePc in the complex) can exist at the same static tip bias depending on the switched configuration, not due to a continuous influence of the tip voltage. Accordingly, we have revised the manuscript in parts so that this misconception is avoided.

This partially answers the second question of the reviewer: We agree with the reviewer that a static finite tip-voltage can influence the spin-orbit coupling (SOC), and we acknowledge that even without a static tip-voltage, SOC will be present in our system. In the spin model, SOC is included effectively through the magnetic anisotropy parameters and the g-factor, consistent with the approach used for isolated Fe atoms in STM studies (see Ref. 55, W. Paul et al.). In

general, we believe that introducing an explicit SOC term does not contribute an additional understanding of the mechanism of the spin switch, since the dominant interactions are exchange coupling and anisotropy. Since SOC is both difficult to obtain experimentally and treat theoretically, we chose to incorporate it in an effective manner, which we find to be both physically meaningful and sufficient for interpreting the key phenomena.

The authors offer several options to discuss inconsistencies between their data and the theoretical predictions e.g. the DFT and IETS and the Resonance frequency shift of FePc target induced by spin switch. I suggest the authors to take into account the role of L – including an electric/spin-Hamiltonian term-and try to clean-up which of these inconsistencies can be fixed.

Reply: As explained above, the SOC term that interacts with the tip as the reviewer suggests is not appropriate, because the state A and B form also in the absence of a static tip-bias. The main inconsistencies between DFT and IETS are the spin state of Fe in the complex, and the sign of the small exchange coupling. We do not see how including the role of L might help. We would like to add that inconsistencies between the experiment and DFT are not surprising: DFT in general has difficulties capturing complex spin configurations, so that the deviations between experiment and theory are not surprising. This was nicely addressed in the comments by reviewer 2.

We believe that the importance of theory in this work is to support the experiments qualitatively to shed light on the mechanisms of both the change in adsorption and in spin state.

Having said that, we have revised the analysis and included the suggestion of the reviewer for the experimental inconsistency of the frequency shift in Supplementary Section 12: Here, we have i) optimized the parameters D and J and ii) included SOC effectively in the g-factor. In this way, we obtain a high population of 90-93% at 300 mK and 0.5 T. Thus, the intensity of the peak that corresponds to the first excited state population becomes negligibly small, which explains why it is not observed in the experiment.

The authors use a simple spin-Hamiltonian

$$H = JS_{\vec{FePc}} \cdot S_{\vec{Fe}} + [D_{FePc} S_{\vec{FePc}}]_{z} (FePc,z)^2 + [D_{Fe} S_{\vec{Fe}}]_{z} (Fe,z)^2 + E_{Fe} (S_{\vec{Fe},x})^2 - S_{\vec{Fe},y}^2$$

to describe their system. This H entails a simple scalar J between $S_{\vec{FePc}}$ and $S_{\vec{Fe}}$ i.e. instead of a J-tensor, as should be in a more complete approach. The authors should justify

this approach. This is critical i.e. the FePc/Fe-atom system on the MgO/Ag(001) plane plus the tip-electric field is a highly-anisotropic system that –in general- is expected to be represented by a non-isotropic-J.

Reply: Again, we would like to clarify that the electric field from the STM tip does not continuously influence the system. The tip voltage is only pulsed briefly to induce switching, and it does not impose a static electric field that would contribute to anisotropic exchange in the spin Hamiltonian. We now comment in the manuscript on the use of a scalar J. While a tensorial J could in principle offer a more realistic description of the exchange interaction in anisotropic environments, we find that a scalar J i) captures all key features observed in our experiments, ii) reduces the complexity of the model, and thus iii) improves the readability. The use of an isotropic J is in-line with most studies in on-surface spin systems, where an isotropic J is applied (e.g. Ref. Zhang et al., Nature Chem. 2022, Choi et al. Nature Nanotechnology 2017 for the scalar J coupling between Fe atoms, FePc molecules and FePc molecules and Ti atoms.).

The DFT calculations are inconclusive in present work. As recognized by the authors in their text DFR failed to predict the Spin-states (if we consider that the experimental IETS are correct). I suggest the authors to remove the DFT part from the manuscript and its – perplexing- discussion with the experiential data. In addition, it is not clear in the description of the DFT-method how the tip-electric-field effect has been taken into account.

Reply: We are convinced that the DFT results are indispensable for this work. Our goal is not to have a perfect agreement between theory and experiment. This is beyond the scope of DFT. Instead, we believe that it is very important to use DFT to reveal the mechanisms by which such a magnetic switch functions. This is important in order to apply the concepts learned here to other spin systems. The key results obtained by DFT are:

1. The atomic-scale positions of Fe and FePc in the two states (Fig. 1e,f)
2. The stability of the states A and B, their difference in energy and how the Fe adatom helps to stabilize the configurations (Fig. 2d)
3. The *mechanism* by which the spin state is changed between state A and B, i.e. the change in population of the ligand spin states that compensates the total spin state (insets in Fig. 3c,f).

Thus, the DFT calculations remain valuable in helping us understand the bistability of the system, the underlying adsorption configurations, and the spin configurations of FePc in both State A and B. We have revised the manuscript in parts to highlight the intentions of the

supporting DFT in gaining a qualitative understanding of the system. However, in our view removing the DFT is not a good idea.

Reviewer #1 (Remarks to the Author):

The authors have well revised the introduction to emphasize the novelty of this report. The referee considers that the manuscript should be published as it is.

Reply: We are grateful to the reviewer for recognizing our work.

Reviewer #2 (Remarks to the Author):

REPORT on the revised version of the paper entitled: " An electrically controlled single molecule spin switch " by W. Huang et al

I am completely satisfied with the revised version of the paper with its annex of supplemental material (SM). The authors answered exhaustively to my comments and questions, clarifying also the origin of the measured Larmor frequency in the target ($S = 1/2$) spin system (mistakenly I had interpreted B_{ext} as the total applied field including the residual tip magnetic field).

In particular I appreciated the new section 3.2 of the SM where the authors discuss the electronic and spin states of the various systems encountered in their work (pristine FePc and complexes A and B) based on DFT calculations. They point out that the main physical effect in these systems is the charge transfer from the substrate to the adsorbed system which is well described by the DFT approach. There is however a further effect not correctly taken into account by this method, which is the description of the multi-configuration interaction of the various excited states of the complex so obtained, which at the end determines its spin state and the correct final amount of the charge transfer. However the authors are aware of this limitation of DFT and "cum grano salis" illustrate the evolution of the physical properties of the various complexes in a convincing way.

I think this paper deserves to be brought to the attention of the general scientific community. (Calogero R. Natoli)

Reply: We sincerely thank the reviewer for the positive evaluation of our revised manuscript and supplementary material. We greatly appreciate the insightful comments and constructive feedback provided during the review process, which have significantly improved the clarity and depth of our work.

Reviewer #3 (Remarks to the Author):

In their Revised manuscript the authors made some text-improvements and provide opinions on my original comments. Although this is a fair response by the esteemed colleague,

I consider that this revised-manuscript lacks the undisputable coherence and cutting-edge conclusiveness, than would qualify it for publication in Nature Communications Journal.

From their replies and the Reviewers comments it is obvious that there are issues and caveats in ESR data analysis (spin states, anisotropy-tensors), as well as in the assumptions needed to be done to accept the inadequacies of DFT to provide all key-parameters. I suggest Decline from publication in Nature Communications Journal.

Reply: We thank the reviewer for their continued assessment of our manuscript and for raising important points regarding the interpretation of the ESR data, the role of spin-orbit coupling (SOC), and the modelling choices in our spin-Hamiltonian approach. We fully acknowledge that there are limitations in both experiment and theory—particularly in the ability of DFT to predict all quantitative parameters in complex spin systems. However, we respectfully disagree with the conclusion that these limitations undermine the coherence or conclusiveness of the work to the extent of warranting rejection.

The focus of our manuscript is the experimental demonstration of a controllable single-molecule spin switch, supported by both ESR and IETS data, and interpreted within a minimal and physically transparent spin model. This combination allows us to clearly identify the change in spin state and magnetic anisotropy upon switching, as well as to directly observe the corresponding effect on a nearby spin target. As stated in our first-round response, the DFT results serve a qualitative role, i.e., elucidating adsorption geometries, state stability, and the spin-switching mechanism, which we consider essential for understanding the system. These theoretical insights complement, but do not compromise, our central findings, which are firmly grounded in our experimental evidence.